# Fcp1 phosphatase controls Greatwall kinase to promote PP2A-B55 activation and mitotic progression

Rosa Della Monica[1,2†], Roberta Visconti[3†], Nando Cervone[1,2], Angela Flavia Serpico[1,2], Domenico Grieco[1,2*]

[1]CEINGE Biotecnologie Avanzate, Naples, Italy; [2]Dipartimento di Medicina molecolare e Biotecnologie mediche, University of Naples Federico II, Naples, Italy; [3]Istituto per l'endocrinologia e l'oncologia "Gaetano Salvatore", Consiglio Nazionale delle Ricerche, Naples, Italy

**Abstract** During cell division, progression through mitosis is driven by a protein phosphorylation wave. This wave namely depends on an activation-inactivation cycle of cyclin B-dependent kinase (Cdk) 1 while activities of major protein phosphatases, like PP1 and PP2A, appear directly or indirectly repressed by Cdk1. However, how Cdk1 inactivation is coordinated with reactivation of major phosphatases at mitosis exit still lacks substantial knowledge. We show here that activation of PP2A-B55, a major mitosis exit phosphatase, required the phosphatase Fcp1 downstream Cdk1 inactivation in human cells. During mitosis exit, Fcp1 bound Greatwall (Gwl), a Cdk1-stimulated kinase that phosphorylates Ensa/ARPP19 and converts these proteins into potent PP2A-B55 inhibitors during mitosis onset, and dephosphorylated it at Cdk1 phosphorylation sites. Fcp1-catalyzed dephosphorylation drastically reduced Gwl kinase activity towards Ensa/ARPP19 promoting PP2A-B55 activation. Thus, Fcp1 coordinates Cdk1 and Gwl inactivation to derepress PP2A-B55, generating a dephosphorylation switch that drives mitosis progression.

*For correspondence: domenico.grieco@unina.it

†These authors contributed equally to this work

Competing interests: The authors declare that no competing interests exist.

## Results and discussion

We recently reported a critical, transcription-independent, role for the essential RNA polymerase II-carboxy-terminal domain (RNAP II-CTD) phosphatase Fcp1 in Cdk1 inactivation at the end of mitosis (*Visconti et al., 2012*). Indeed, depleting Fcp1 from living HeLa cells as well as from mitotic HeLa cell extracts, that are nuclei-free thus non-transcribing, substantially impaired cyclin B degradation, Cdk1 inactivation and mitosis exit (*Visconti et al., 2012*). In that study, we also noticed that Fcp1 depletion impaired bulk mitotic protein dephosphorylation also upon chemical inhibition of Cdk1 in non-transcribing mitotic cell extracts (*Visconti et al., 2012*). This observation suggested that Fcp1 was required for crucial mitosis exit dephosphorylations even downstream Cdk1 inactivation in a transcription-independent manner. However, bulk dephosphorylation at mitosis exit are likely due to action of major phosphatases like PP1 or PP2A, rather than Fcp1 itself (*Ferrigno et al., 1993*; *Qian et al., 2013*). The PP2A-B55 isoform, in particular, has relevant roles for late mitotic events like spindle breakdown, chromatin decondensation, nuclear membrane and Golgi reassembly and cytokinesis (*Schmitz et al., 2010*; *Cundell et al., 2013*). In addition, PP2A-B55 has been shown to reverse bulk mitotic phosphorylations detectable by a commercially available anti-Cdk1 substrate antibody, recognizing the K/HpSP motif (where pS is phosphorylated Ser), and the phosphorylation of PRC1, a crucial cytokinesis protein, at T481 (*Schmitz et al., 2010*; *Cundell et al., 2013*; *Qian et al., 2013*). In preliminary experiments, in which Fcp1 expression was downregulated in HeLa cells by small interfering RNAs (siRNAs), we found that dephosphorylations of bulk K/HpSP motif and pT481-PRC1 at

**eLife digest** Cells multiply through a cell division cycle that has distinct phases. In a phase called mitosis, a cell splits its genetic material, which was duplicated in a preceding phase, into two identical sets. Each of these sets will form the genetic material of daughter cells. If this process goes wrong, then cells can die or become cancerous, and so cells have evolved a complex regulatory process to ensure that mitosis begins and ends at the correct time.

For mitosis to begin, an enzyme adds tags called phosphate groups to hundreds of target proteins. These phosphate groups are then removed again to end mitosis. PP2A-B55 is an enzyme that removes these phosphate groups and is needed to complete mitosis, but must remain inactive before this point. This inactivation occurs because a protein called Greatwall activates two other proteins that inhibit PP2A-B55. To reactivate PP2A-B55 at the end of mitosis, Greatwall must be inactivated, but it was not known how cells do this.

Della Monica, Visconti et al. have now investigated this process in human cells. The experiments show that towards the end of mitosis, another enzyme called Fcp1 inactivates Greatwall by removing phosphate groups from it. This allows PP2A-B55 to reactivate.

These studies reveal that Fcp1 is a key factor that is needed to complete mitosis. The next challenge is to determine how Fcp1 activity is regulated at the end of mitosis.

mitosis exit were indeed dependent on Fcp1 (*Figure 1—figure supplement 1*). However, given the role for Fcp1 in inactivation of the spindle assembly checkpoint and of Cdk1 (*Visconti et al., 2012*; *Visconti et al., 2013*), delayed dephosphorylations could be due to persistence of Cdk1 kinase activity rather than impaired PP2A-B55 phosphatase activation at the end of mitosis. To know whether Fcp1 controlled PP2A-B55 activation downstream Cdk1 inactivation, we determined whether Fcp1 depletion impaired bulk K/HpSP motif and pT481-PRC1 dephosphorylation upon chemical inhibition of Cdk1 activity in mitotic cells and cell extracts. Control and Fcp1 siRNAs-depleted, as well as Fcp1 depleted complemented with siRNAs-resistant wild type Fcp1 (Fcp1WT) expression vector, HeLa cells were arrested at pro-metaphase and further treated with the Cdk1 inhibitor RO-3306 (*Figures 1A,B*). Nuclei-free, mitotic HeLa cell extracts were, instead, either mock immunodepleted, as control, or immunodepleted of Fcp1 or immunodepleted of Fcp1 and reconstituted with purified, active, Fcp1 wild type (Fcp1WT) protein before treatment with RO-3306 (*Figures 1C,D*). The results showed that, in cells and cell extracts, Fcp1 was indeed required for timely PP2A-B55-dependent dephosphorylations following Cdk1 inactivation (*Figure 1A,C*). Fcp1 is relatively resistant to the potent PP2A inhibitor okadaic acid (OA); nevertheless, we confirmed that pT481-PRC1 and bulk K/HpSP motif dephosphorylations were OA sensitive (*Figure 1—figure supplement 2*) (*Schmitz et al., 2010*; *Cundell et al., 2013*).

Considering that pT481-PRC1 and bulk K/HpSP motif dephosphorylations at mitosis exit have been shown to depend on PP2A-B55 activity (*Schmitz et al., 2010*; *Cundell et al., 2013*), together, our data suggested that PP2A-B55 activation might require Fcp1 downstream Cdk1 inactivation.

Whether pT481-PRC1 dephosphorylation is necessary for PRC1 localization and spindle midzone organization is debated (*Cundell et al., 2013*; *Hu et al., 2012*). Nevertheless, we found that in Fcp1-depleted cells PRC1 poorly concentrated at spindle midzone, even upon Cdk1 inhibition (*Figure 1—figure supplements 3A,3B*), and that Fcp1 depletion induced several mitotic phenotypes in asynchronously growing cells, in addition to cell death, including accumulation of binucleated cells, suggesting that Fcp1 was indeed required also for proficient cytokinesis (*Figure 1—figure supplement 4*).

During mitosis onset, PP2A-B55 is inhibited by a recently elucidated pathway: Cdk1 phosphorylates and stimulates Gwl kinase that, in turn, represses PP2A-B55 activity by phosphorylating Ensa/ARPP19 and converting these proteins into potent PP2A-B55 inhibitors (*Schmitz et al., 2010*; *Castilho et al., 2009*; *Vigneron et al. 2009*; *Lorca and Castro, 2013*; *Mochida and Hunt, 2012*). How this condition is reversed at the end of mitosis is still unclear.

The Fcp1 phosphatase has already been called in the mechanism for PP2A-B55 reactivation at mitosis exit (*Hegarat et al., 2014*). Indeed, Fcp1-depleted, mitotic cells were found to maintain

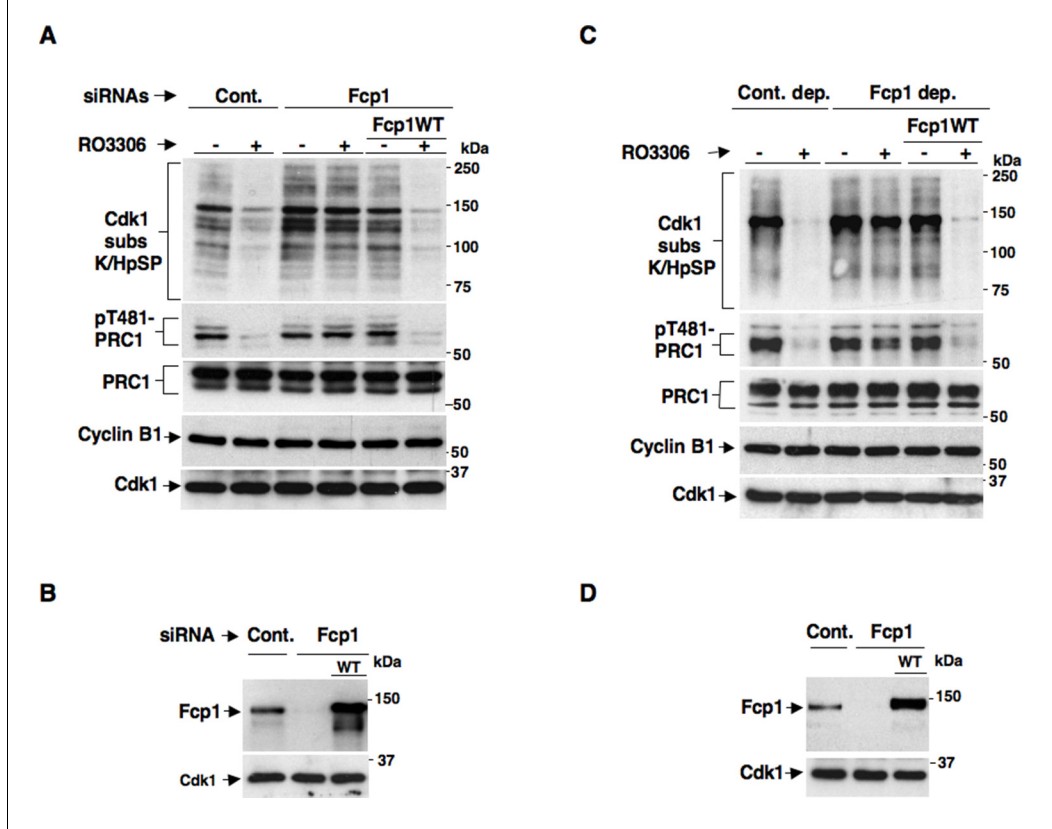

**Figure 1.** Fcp1 affects PP2A-B55-dependent dephosphorylations at mitosis exit. (A, B) Control (Cont.) or Fcp1-depleted (Fcp1) by siRNAs, as well as Fcp1-depleted complemented with wild type Fcp1 (Fcp1WT), HeLa cells were arrested at prometaphase. (A) Cells were collected and split into two samples, one sample received vehicle (-), the other RO3306 (+), and they were further incubated for 15 min, lysed and lysates probed for indicated antigens. (B) Cells were lysed and lysates probed for indicated antigens. (C, D) Mitotic HeLa cell extracts were mock-depleted (Cont. dep.), Fcp1-depleted (Fcp1 dep.) or Fcp1-depleted and reconsituted with Fcp1WT. (C) Each set was split into two portions, one received just vehicle (-), the other RO3306 (+), and they were incubated for 30 min at 23°C and probed for indicated antigens. (D) Before incubation, extracts samples were also probed for Fcp1 and Cdk1. The data shown are representative of three independent experiments per type.

The following figure supplements are available for figure 1:

**Figure supplement 1.** Fcp1-dependency of mitotic exit dephosphorylations.

**Figure supplement 2.** Okadaic acid-sensitive mitotic exit dephosphorylations.

**Figure supplement 3.** PRC1 localization in Fcp1-depleted cells.

**Figure supplement 4.** Mitotic phenotypes in Fcp1-depleted cells.

Gwl-dependent phosphorylation of Ensa (at S67 in human Ensa; pS67-Ensa) upon Cdk1 inhibition, and some evidence was provided that Fcp1 directly dephosphorylated pS67-Ensa to allow PP2A-B55 reactivation (*Hegarat et al., 2014*). However, this evidence was subsequently challenged by Williams and co-workers who showed that PP2A-B55 itself, rather than Fcp1, dephosphorylated pS67-Ensa and autoactivated, although this slow reaction could be rapidly reversed as long as Gwl stays active (*Williams et al., 2014*). We tested the ability of Fcp1 to dephosphorylate specifically pS67-Ensa in vitro and found that, in agreement with Williams and coworkers conclusions (*Williams et al., 2014*), Fcp1 was unable to do so (*Figure 2A*).

To address mechanistically the Fcp1 dependency of PP2A-B55 activation, we hypothesized that Fcp1 was required to inactivate the Ensa/ARPP19 kinase ability of Gwl and allow PP2A-B55 autoactivation. Gwl is higly phosphorylated in mitosis and several observations indicate that Cdk1 directly

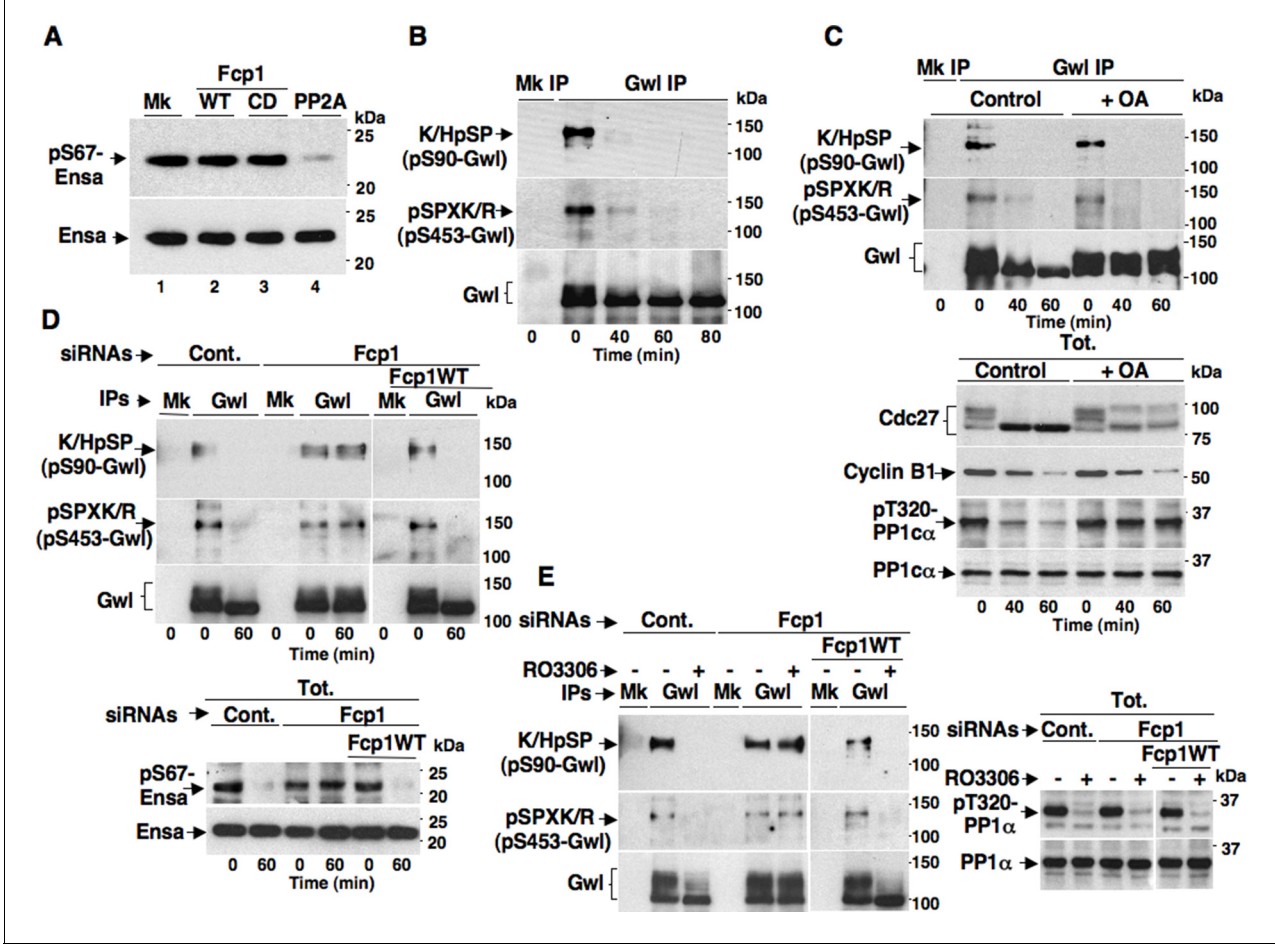

**Figure 2.** Fcp1-dependency of Gwl dephosphorylation at S90 and S453 during mitosis exit. (**A**) Ensa IP from prometaphase-arrested HeLa cells, previously transfected with a Flag-hEnsa expression vector, was divided into four sets and incubated as substrate in phosphatase reactions with either just buffer (Mk) or with purified active Fcp1 (WT), or an inactive, catalytic dead, Fcp1 version (CD), or active PP2A for 60 min at 30°C (lanes 1, 2, 3 and 4, respectively). The reactions were then probed for pS67Ensa and Ensa content (the Fcp1WT and Fcp1CD protein preparations were the same used in the experiment depicted in *Figure 3C*; see below). (**B**) Prometaphase-arrested HeLa cells were released into fresh medium and sampled at indicated time points of further incubation, lysed and processed for mock (Mk) or Gwl IP. IPs were separated in parallel SDS/PAGE and probed for indicated antigens. (**C**) Prometaphase-arrested HeLa cells were released into fresh medium and split into two portions; then, immediately after taking the time 0 sample, vehicle (Control) or 2 µM okadaic acid (+ OA) were added. Cells were further sampled at indicated time points of incubation. Upper panels, cell lysates were processed for mock (Mk) or Gwl IP that were probed for indicated antigens; lower panels, total cell lysate samples (Tot.) were probed for indicated antigens. (**D**) Control (Cont.) or Fcp1-depleted (Fcp1), as well as Fcp1-depleted complemented with wild type Fcp1 (Fcp1WT), HeLa cells were arrested at prometaphase, released and sampled at the indicated time points of incubation. Upper panels, cell lysates were processed for mock (Mk) or Gwl IP that were probed for indicated antigens; lower panels, total cell lysate samples (Tot.) were probed for indicated antigens. (**E**) Control (Cont.) or Fcp1-depleted (Fcp1), as well as Fcp1-depleted complemented with wild type Fcp1 (Fcp1WT), HeLa cells were arrested at prometaphase and split into two samples, one sample received vehicle (-) the other RO3306 (+), further incubated for 15 min. Left panels, cell lysates were processed for mock (Mk) or Gwl IP that were probed for indicated antigens; right panels, total cell lysate samples (Tot.) were probed for indicated antigens. The levels of Fcp1 depletion and complementation were similar to those shown in *Figure 1B*. Data shown are representative of at least four independent experiments per type.

The following figure supplements are available for figure 2:

**Figure supplement 1.** Cdk1-dependent phosphorylations of Gwl.

**Figure supplement 2.** Effect of prolonged Cdk1 inhibition on Gwl dephosphorylation in Fcp1-depleted cells.

phosphorylates Gwl stimulating its kinase activity (*Vigneron et al., 2011*; *Blake-Hodek et al., 2012*; *Dephoure et al., 2008*). Once activated by Cdk1, Gwl can autophosphorylate and autophosphorylation appears to contribute to its own overall activity (*Blake-Hodek et al. 2012*). A study in *Xenopus laevis* egg extracts has very recently provided compelling evidence that PP1 is the phosphatase that dephosphorylates Gwl at autophosphorylation sites, contributing this way to Gwl inactivation at the end of mitosis. However, the same study showed that dephosphorylation of Gwl at other sites, most likely also those phosphorylated by Cdk1, is PP1-independent (*Heim et al., 2015*). As Fcp1 is known to be able to reverse Cdk-dependent phosphorylation (*Ghosh et al., 2008*; *Visconti et al., 2012*), we hypothesised that Fcp1 was required for Gwl inactivation by removing Cdk1-dependent, activatory, phosphorylations of Gwl at the end of mitosis. First, we established a way to monitor changes at potential Cdk1-dependent Gwl phosphorylation sites during mitosis exit. Two commercially available anti-Cdk1 substrate antibodies, the previously mentioned anti-K/HpSP motif and an anti-pSPXR/K (where pS is phosphorylated serine and X any aminoacid) motif, can in principle recognize serine 90 and serine 453 in human Gwl (S90-Gwl and S453-Gwl), respectively, being S90-Gwl (89-KSP-91) and S453-Gwl (453-SPCK-456), the only human Gwl serine residues in those contexts. While S453-Gwl has been shown to be specifically phosphorylated in mitosis by proteomic approaches, S90-Gwl has not (*Dephoure et al., 2008*; *Blake-Hodek et al., 2012*). Nevertheless, both antibodies reacted against V5-tagged wild-type Gwl (V5-GwlWT) but not against V5-Gwl versions in which serine 90 and serine 453 were respectively mutated into non-phosphorylatable alanine (V5-GwlS90A; V5-GwlS453A) when the tagged proteins were isolated from transfected, mitotic, HeLa cells, indicating that both residues are phosphorylated in mitosis (*Figure 2—figure supplement 1A*). In addition, these antibodies did not react against V5-GwlWT isolated from HeLa cells in G1, unless it was treated with purified, active, Cdk1 in vitro (*Figure 2—figure supplement 1B*). To analyse potential changes in Gwl phosphorylation at S90 and S453 during mitosis exit, we probed endogenous Gwl isolated from HeLa cells taken at various time points during mitosis exit with anti-K/HpSP and pSPXR/K antibodies (*Figure 2B*). PS90- and pS453-Gwl signals were readily detected in prometaphase but were progressively lost as cells transited out of mitosis (*Figure 2B*). Importantly, dephosphorylation at both sites was resistant to OA at a dose (2 μM) that potently inhibited PP1 (*Figure 2C*), as indicated by persistence, despite Cdk1 inactivation by cyclin B degradation, of Cdk1-dependent inhibitory phosphorylation of PP1 catalytic subunit a (PP1cα) at T320 (pT320-PP1cα), a site that PP1 autodephosphorylates upon Cdk1 inactivation (*Qian et al., 2013*; *Heim et al., 2015*) (*Figure 2C*). However, OA significantly delayed the kinetics of Gwl migration shift on SDS/PAGE, in agreement with the notion that also OA-sensitive phosphatase(s) dephosphorylate Gwl at several other sites during mitosis exit (*Hegarat et al., 2014*; *Williams et al., 2014*; *Heim et al., 2015*). Conversely, depletion of Fcp1 delayed Gwl dephosphorylation at both S90 and S453 as well as Gwl migration shift and pS67-Ensa dephosphorylation (*Figure 2D*). In addition, a 15-min treatment with RO-3306 promptly induced pS90- and pS453-Gwl dephosphorylation and Gwl downshift in prometaphase-arrested control and Fcp1 re-expressing cells but not in prometaphase-arrested Fcp1-depleted cells, indicating that Fcp1 was required for these dephosphorylations downstream Cdk1 inactivation (*Figure 2E*). Prolonging Cdk1 inhibitor treatment up to 30 min resulted in some Gwl dephosphorylation also in Fcp1 siRNAs-treated cells (*Figure 2—figure supplement 2*); however, we could not rule out whether this was caused by the action of other phosphatases or of residual Fcp1 after substantial time from Cdk1 inactivation. Nevertheless, the 15-min treatment with RO-3306 was able to potently induce autoactivatory pT320-PP1cα dephosphorylation in Fcp1-depleted as in control cells (*Figure 2E*). Thus, Fcp1 is required for timely dephosphorylation of at least two Cdk1-dependent sites of Gwl, S90 and S453, at mitosis exit.

We set out to determine whether Fcp1 directly dephosphorylated Gwl at S90 and S453. As Fcp1 can be found in complexes with its substrates (*Visconti et al., 2015*), we first asked whether Fcp1 and Gwl physically interacted during mitosis exit. Indeed, by co-immunoprecipitation (coIP) of endogenous proteins, we found that Gwl transiently bound Fcp1 during mitosis exit (*Figure 3A*). Binding was lower in prometaphase, increased during the period of spindle assembly (20-30 min) to decrease thereafter (*Figure 3A*; Fcp1 and Gwl appear to be similarly abundant proteins in HeLa cells and we estimate that approximately 8-14% of Fcp1 interacts with Gwl in cells at the peak of interaction as, routinely, ~3-5% of total lysate Fcp1 was found in Gwl IP that contained ~30-40% of total lysate Gwl). In Fcp1-depleted cells complemented with exogenous Fcp1, Gwl interacted with the exogenous protein with similar kinetics observed with the endogenous Fcp1 (*Figure 3B*). Transient

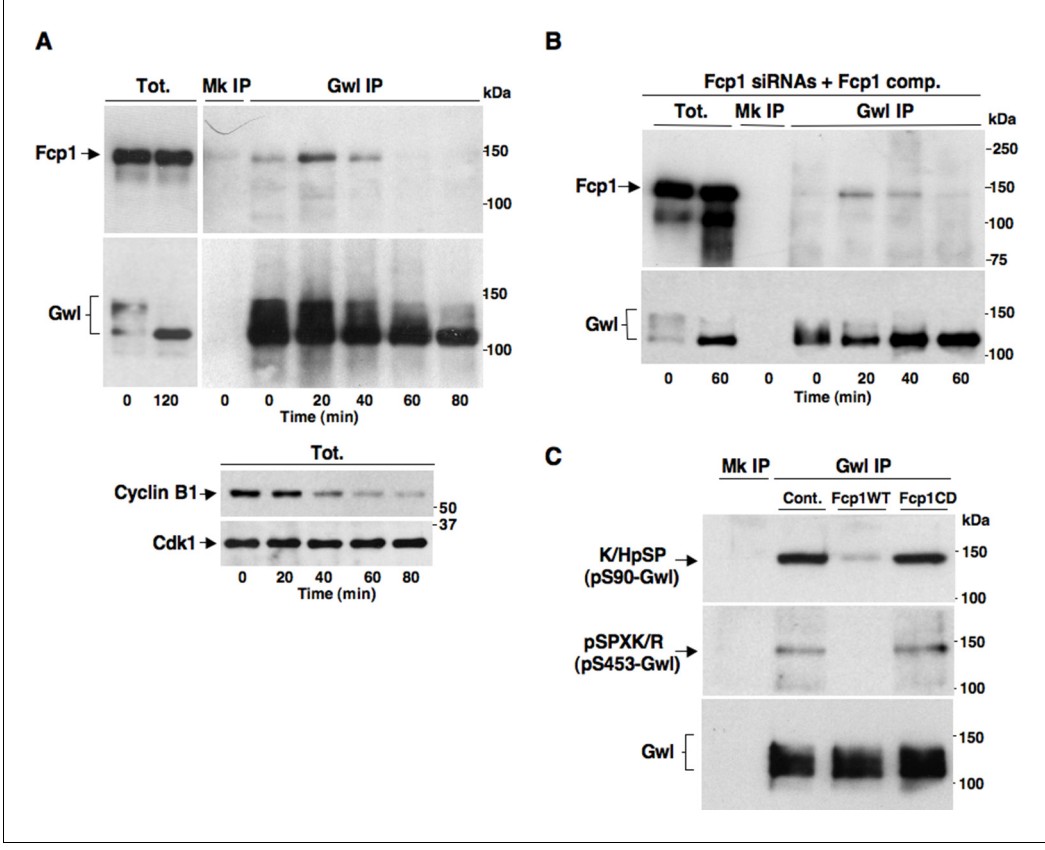

**Figure 3.** Fcp1 binds and dephosphorylates Gwl during mitosis exit. (**A**) Prometaphase-arrested HeLa cells were released into fresh medium and sampled at indicated time points of incubation. Upper panels, cell lysates were processed for mock (Mk) or Gwl IP that were probed for indicated antigens; lower panels, total cell lysate samples (Tot.) were probed for indicated antigens. (**B**) Prometaphase-arrested, Fcp1-siRNAs-transfected and complemented with siRNAs-resistant Fcp1WT (Fcp1-siRNAs +Fcp1 comp.) HeLa cells were released into fresh medium and sampled at indicated time points of incubation. Total lysates (Tot.), mock (Mk) or Gwl IPs were probed for indicated antigens. Data shown are representative of at least three independent experiments per type. (**C**) Gwl IP from prometaphase-arrested HeLa cell lysates was divided into three sets and incubated in phosphatase reactions with either just buffer (Cont.), Fcp1WT or Fcp1CD proteins (Mk IP; 1/3 mock IP incubated with buffer) for 60 min at 30°C. Then, IPs were washed and probed for indicated antigens. Data shown are representative of three independent experiments per type.

The following figure supplement is available for figure 3:

**Figure supplement 1.** Fcp1-Gwl interaction in hTERT-RPE1 cells.

Gwl-Fcp1 interaction was also detected during mitosis exit in non-transformed, telomerase-immortalized, human retinal pigment epithelium cells hTERT-RPE1 (*Figure 3—figure supplement 1*).

Taken together, these data suggest that Fcp1 bound and dephosphorylated Gwl at S90 and S453, and possibly at other Cdk1-dependent sites, during mitosis exit and that Fcp1-catalyzed dephosphorylation lowered Gwl activity towards Ensa/ARPP19, allowing PP2A-B55 to autoactivate.

To investigate this hypothesis, we first asked whether purified Fcp1 could dephosphorylate Gwl at S90 and S453 in vitro. Indeed, purified active Fcp1 (Fcp1WT), but not an inactive, catalytic dead, Fcp1 version (Fcp1CD), was able to dephosphorylate Gwl isolated from mitotic cells at both sites (*Figure 3C*). Next, we asked whether Fcp1-dependent dephosphorylation of Gwl in vitro lowered its kinase activity towards Ensa/ARPP19. To this end, endogenous Gwl, isolated from mitotic cells, was mock-treated or treated with active Fcp1WT or inactive Fcp1CD proteins as described in *Figure 3C*. Subsequently, Gwl activity was assessed on recombinant *X. laevis* Ensa protein as substrate (*Figure 4A*). The results clearly show that pre-treatment of mitotic Gwl with Fcp1WT, but not with

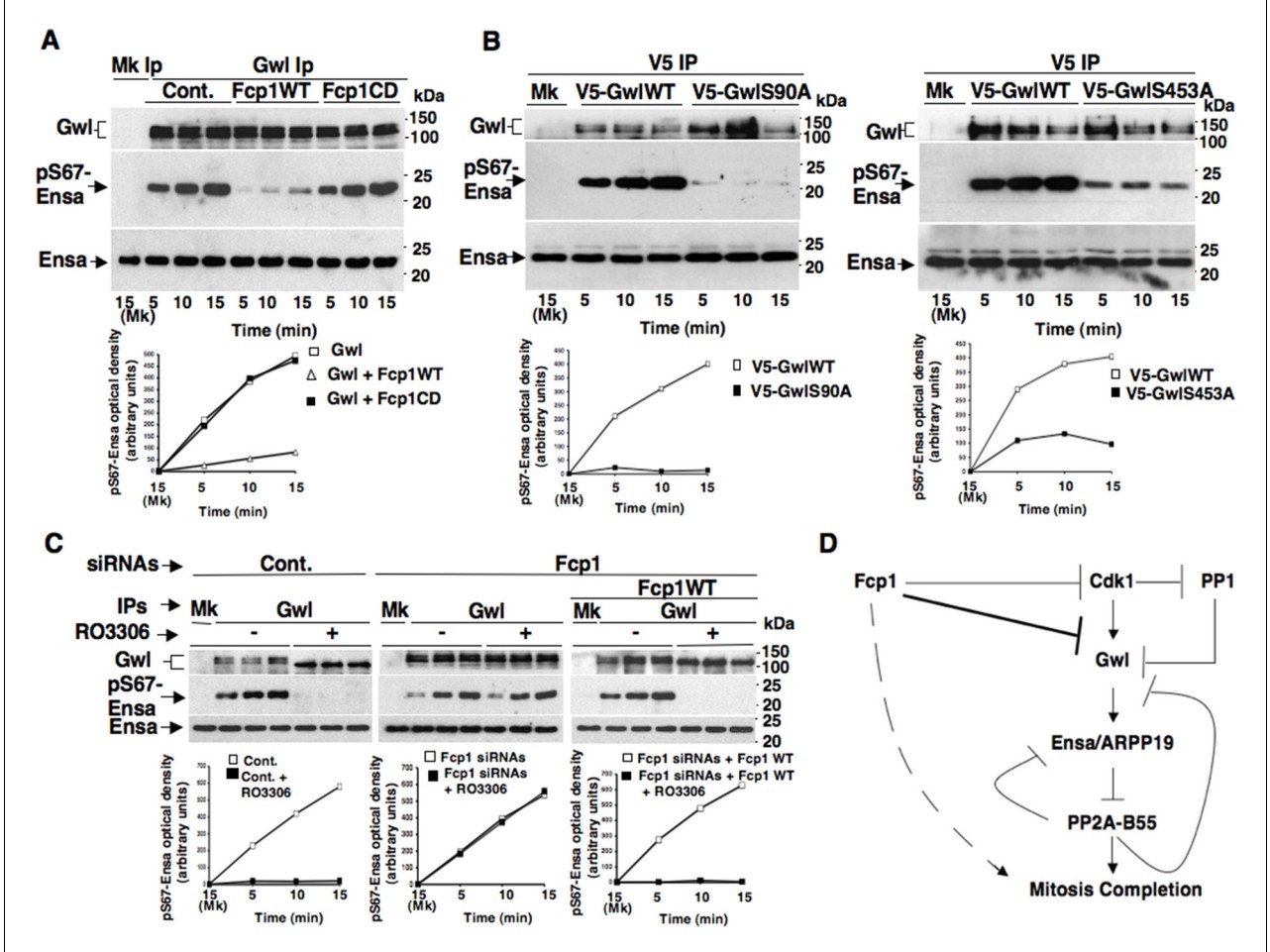

**Figure 4.** Fcp1-dependent dephosphorylation reduces Gwl kinase activity towards Ensa. (**A**) Gwl IP from prometaphase-arrested HeLa cell lysates were divided into three sets and incubated for phosphatase reactions with either just buffer (Cont.), Fcp1WT or Fcp1CD proteins (Mk IP; 1/3 mock IP incubated with buffer) for 60 min at 30°C. Then, each IP set was split into three portions and incubated in kinase reactions with recombinant Ensa protein. (**B**) V5 IP from lysates of prometaphase-arrested HeLa cells, previously transfected with V5-GwlWT and V5-GwlS90A or V5-GwlS453A, were split into three portions and incubated at 37°C in kinase reactions with recombinant Ensa protein. (**C**) Gwl IPs from the same cell lysates of the experiment described in *Figure 2E*, in which prometaphase-arrested control (Cont.) and Fcp1 siRNAs-treated (Fcp1), as well as Fcp1 siRNAs-treated complemented with wild type Fcp1 (Fcp1WT), HeLa cells were treated – or + RO3306 for 15 min, split into three portions and incubated in kinase reactions at 37°C and with recombinant Ensa protein. Kinase reactions were stopped at indicated time points of incubation and probed for indicated antigens (Mk IPs were incubated for 15 min). Graphs show quantitation (arbitrary units) of pS67-Ensa optical density. Data shown are representative of three independent experiments per type. (**D**) Schematic for the proposed model of PP2A-B55 control.

The following figure supplements are available for figure 4:

**Figure supplement 1.** Fcp1 affects Ensa/ARPP19 kinase ability of Gwl.

**Figure supplement 2.** Effects of prolonged Cdk1 inhibition on Gwl activity in Fcp1-depleted cells.

Fcp1CD, substantially reduced Gwl ability to phosphorylate S67-Ensa (*Figure 4A*). Similar results were obtained with V5-GwlWT isolated from transfected, mitotic, cells or using ARPP19 as Gwl substrate (*Figure 4—figure supplement 1A,1B*). We cannot exclude that, in addition to S90 and S453, other Cdk1 phosphorylation sites in Gwl are dephosphorylated by Fcp1; nevertheless, assaying S67-Ensa kinase activity of V5-GwlS90A and V5-GwlS453A mutant proteins, isolated from transfected and prometaphase-arrested HeLa cells, revealed that both mutants had significantly reduced S67-Ensa kinase activity compared to V5-GwlWT (*Figure 4B*). The S90A mutation of human Gwl showed highest kinase reduction similarly to what reported for a non-phosphorylatable mutant at equivalent

residues in *X. laevis* Gwl (*Blake-Hodek et al., 2012*). Moreover, a non-phosphorylatable Gwl mutant at S452 (V5-GwlS452A), just preceding serine S453, that is not in a Cdk1 phosphorylation consensus but is found phosphorylated in mitotic cells (*Dephoure et al., 2008*), had S67-Ensa kinase activity similar to GwlWT (*Figure 4—figure supplement 1C*). In addition, we assessed the S67-Ensa kinase activity of Gwl isolated from the same lysates of the experiment described in *Figure 2E* in which control, Fcp1-depleted and Fcp1-depleted and complemented prometaphase-arrested cells were treated with the Cdk1 inhibitor for 15 min, and found that while this treatment completely abolished Gwl activity in control cells it had negligible effects in Fcp1-depleted cells (*Figure 4C*; see also *Figure 2E*). Like for Gwl dephosphorylation, prolonging Cdk1 inhibitor treatment up to 30 min resulted in a reduction of Ensa kinase activity of Gwl also in Fcp1-depleted cells and, again, we cannot rule out whether this is due to the fact that other phosphatases or residual Fcp1 dampened Gwl activity after substantial time from Cdk1 inactivation (*Figure 4—figure supplement 2*). Nevertheless, it is important to remark that the 15-min treatment with RO-3306 induced full pT320-PP1cα dephosphorylation in Fcp1-depleted cells without significantly affecting Gwl kinase activity towards Ensa (*Figures 2E* and *4C*).

Together, these data indicate that Fcp1-dependent dephosphorylation greatly reduces S67-Ensa kinase activity of Gwl and that, downstream inactivation of Cdk1, Fcp1 deficit substantially blunts inactivation of Gwl.

We previously reported that the Fcp1 phosphatase is required to perform dephosphorylations that ultimately bring about Cdk1 inactivation at the end of mitosis (*Visconti et al., 2012*). We report now that, downstream Cdk1 inactivation, Fcp1 dephosphorylates Gwl at sites probably phosphorylated by Cdk1 and downregulates Gwl kinase activity towards Ensa/ARPP19 (*Figure 4D*). This ultimately leads PP2A-B55 to take the upper hand in dephosphorylating and releasing Ensa/ARPP19 as competitive inhibitors, getting free to dephosphorylate other substrates to complete mitosis (*Williams et al., 2014*). Recently, PP1 has been involved in the mechanism of Gwl inactivation at the end of mitosis by reversing Gwl activatory autophosphorylation (*Heim et al., 2015*). We found that, upon Cdk1 inactivation, Gwl inactivation is strongly delayed if Fcp1 is downregulated, despite potentially active PP1 (see *Figures 2E* and *4C*). As Cdk1-dependent phosphorylation stimulates Gwl activity not only towards Ensa/ARPP19 but also towards Gwl itself at autoactivatory sites (*Blake-Hodek et al., 2012*), by reversing Cdk1-dependent phosphorylation Fcp1 could also reduce Gwl autoactivatory strength, favouring this way PP1 action to stably switch off Gwl autoactivation and, along with directly reducing Gwl activity towards Ensa/ARPP19, allow PP1 and PP2A-B55 to shut Gwl activity off definitively (*Hegarat et al., 2014*; *Williams et al., 2014*; *Heim et al., 2015*). Thus, by controlling Cdk1 and Gwl inactivation, Fcp1 appears to be at the apex of a phosphorylation cascade required to exit mitosis (*Figure 4D*). Along with other recently described phosphatase activation networks (*Lorca and Castro, 2013*; *Porter et al., 2013*; *Nijenhuis et al., 2014*; *Grallert et al., 2015*), this pathway contributes to ensure coordinated reversal of mitotic phosphorylations to grant correct completion of mitosis.

# Materials and methods

## Cell culture and treatments

HeLa and hTERT-RPE1 cells were grown and maintained as previously described (*Visconti et al., 2012*; *Visconti et al., 2010*). Prometaphase-arrested cells were obtained by performing a double thymidine (4 mM; Sigma-Aldrich, St. Louis, MO) block (18 hr each, separated by a 6 hr incubation in fresh medium) followed by release into fresh medium containing nocodazole (500 nM; Calbiochem, Billerica, MA) and incubation for 12 or 14 hr for HeLa and hTERT-RPE1, respectively. Release from prometaphase arrest was obtained by washing detached cells twice with PBS and twice with fresh medium, followed by incubation in fresh medium. Cells in G1 were obtained after 120 min incubation from prometaphase release.

For asynchronous siRNAs treatment, Hela cells were transfected with non-targeting or human Fcp1 3′ UTR-targeting (5′-guaagugacagguguuaaa-3′) siRNAs (Dharmacon Inc., Lafayette, CO). For siRNAs treatment and complementation experiments, HeLa cells were first transfected with 3XFlag-Fcp1WT expression vector (or empty vector for mock transfections; *Visconti et al., 2012*). Eight hours post transfection, cells were treated with thymidine (4 mM; Sigma-Aldrich) for 18 hr. Cells

were released from thymidine block into fresh medium and transfected with non-targeting or human Fcp1-targeting siRNAs as above. Cells were treated again with thymidine 6 hr after siRNAs transfections, and incubated for further 18 hr. Cells were then released from the second thymidine block into fresh medium containing nocodazole (500 nM; Calbiochem) for 12 hr. Mitotic extracts from prometaphase-arrested HeLa cells (checkpoint extracts) were produced, Fcp1-immunodepleted and complementated exactly as previously described (*Visconti et al., 2012*). Recombinat Fcp1WT and Fcp1CD proteins were produced and stored in EXB (20 mM HEPES pH 7.6, 5 mM KCl, 1mM DTT, 100 µg/ml 3XFLAG peptide, 10% glycerol; Sigma-Aldrich) as previously described (*Visconti et al., 2012*). V5-GwlWT expression vector was obtained by subcloning pENTR221-Gwl clone into V5-tagged pcDNA3.1 vector (Invitrogen, Carlsbad, CA). To generate the V5-Gwl-S90A mutant, V5-Gwl-S453A mutant and V5-Gwl-S452A mutant, serine 90, serine 453 and serine 452 of human Gwl were mutagenized into alanine by QuikChange II XL site-directed mutagenesis kit (Agilent Technologies, Santa Clara, CA) using the V5-GwlWT expression construct as template. 3XFlag-Fcp1WT or 3XFlag-Fcp1CD expression vectors have been previously described (*Visconti et al., 2012*). Flag-hEnsa expression vector was purchased from Origene (Rockville, MD). Eukaryotic expression vectors transfections were performed using Linear Polyethlenimine (Polysciences Inc., Warrington, PA). Recombinant 6His-tagged *X. laevis* Ensa and ARPP19 proteins were expressed in BL21 *E. coli* cells and purified using Ni-NTA agarose kit (Qiagen, Germany) according to the manufacturer's instructions. The Cdk1 inhibitor RO3306 (Calbiochem) was used at 5 and 50 µM in cells and mitotic cell extracts, respectively. Okadaic acid (Calbiochem) was used at 500 nM or 2 µM as indicated, MG132 (Calbiochem) at 10 µM. Cell viability was assessed by trypan blue (EuroClone, Italy) exclusion and immunoblots for cleaved caspase 3.

## Immunological procedures

Anti-phospho-serine Cdk Substrates (P-S2-100; recognizing K/HpSP), anti phosphorylated MAPK/CDK substrates (recognizing PXpSP or pSPXK/R) and anti phospho-ENSA/ARPP19 (pS67/pS62), anti phosphoT320-PP1cα and Cleaved Caspase-3 antibodies were purchased from Cell Signaling Technology (Danvers, MA); anti-MASTL antibodies from Bethyl Laboratories (Montgomery, TX) and NOVUS (Littleton, CO); anti-Fcp1 antibodies from Bethyl Laboratories and Santa Cruz Biotechnology (Dallas, TX). Other antibodies were from Santa Cruz Biotechnology. Immunoprecipitations and immunoblots were performed as previously described (*Visconti et al., 2012*). For immunofluorescence, cells were grown or spun on microscopy slides, washed in PBS, fixed with 4% formaldehyde in PBS for 10 min and permeabilized with 0.2% Triton X-100 in PBS for further 10 min. After blocking with 3% BSA in PBS for 1 hr, samples were incubated with primary antibodies in PBS + 1% BSA for 3 hr. After 3 PBS washes, samples were incubated with secondary antibodies (Jackson ImmunoResearch Laboratories Inc., Westgrove, PA) in PBS + 1% BSA for 1 hr at room temperature. DNA was stained by incubation with Hoechst 33258 (10 µg/ml; Santa Cruz Biotechnology) in PBS. Samples were observed and photographed using an Axiovert 200M inverted microscope equipped with the Apotome slider module with 63X or 40X objectives (Zeiss, Germany).

## In vitro treatments and assays

For in vitro dephosphorylation assays, endogenous Gwl IP or V5 IP or Flag IP, from previously V5-GwlWT- or Flag-hEnsa-transfected cells respectively, from 3 ml lysates, of ~1.5 mg/ml of protein concentration, of prometaphase-arrested cells were washed in phosphatase assay buffer (PAB: 20 mM HEPES, pH 7.6, 10 mM MgCl$_2$, 1 mM dithiothreitol), split into three portions, each containing approximately 500 ng of Gwl, and incubated at 30°C for 1 hr in 10 µl of either PAB + 1/10 volume of EXB, as control, PAB + 1/10 volume of Fcp1WT (50 ng/µl; final protein conc.) or PAB + 1/10 volume of Fcp1CD (50 ng/µl; final protein conc.) or PAB + 1/10 volume of purified PP2A (0.1 unit per reaction; Merck Millipore, Billerica, MA). After phosphatase reaction, samples were separated on SDS/PAGE and probed for the indicated antigens or, where indicated, further processed for Gwl kinase activity assays. For Gwl kinase, after phosphatase reactions, each IP set was washed with EB (80 mM β-glycerophosphate, 10 mM MgCl$_2$ and 20 mM EGTA), divided into three aliquots and incubated for indicated time points in EB buffer supplemented with 1 mM ATP, 10 mM phosphocreatine, 0.1 mg/ml creatine phosphokinase (kinase buffer, KB) and recombinant *X. laevis*, Ensa or ARPP19 proteins (1 µg per sample). One-tenth of each reaction was probed on separate blots for total Ensa or ARPP19

proteins, the remaining was probed for pS67/62-Ensa/ARPP19 and Gwl. Total Ensa was also visualized by re-probing blots previously probed for pS67-Ensa. For in vitro Gwl rephosphorylation assays, V5 IPs from mock- or V5-GwlWT-transfected, G1-synchronised, HeLa cells were split in two portion and incubated at 37°C for 20 min in KB − or + active Cyclin A2-CDK1 (126 ng/reaction, ProQuinase, Germany).

## Acknowledgements

The authors wish to thank VE Avvedimento and RM Melillo for helpful suggestions, S Mochida for providing *X. laevis* ENSA and ARPP19 expression vectors. Supported by a grant of Associazione Italiana per la Ricerca sul Cancro (AIRC) N. IG 2014 Id.15476 to DG.

## Additional information

### Funding

| Funder | Grant reference number | Author |
| --- | --- | --- |
| Associazione Italiana per la Ricerca sul Cancro | IG 2014 Id.15476 | Domenico Grieco |

The funders had no role in study design, data collection and interpretation, or the decision to submit the work for publication

### Author contributions

RDM, Made initial observations on the Fcp1-Gwl interaction and designed experiments. Performed IP/blot experiments. Performed subcloning and site directed mutagenesis. Performed phosphatase and kinase assays. Analysed and discussed all data.; RV, Made initial observations on the Fcp1-Gwl interaction and designed experiments. Performed IP/blot experiments. Performed subcloning and site directed mutagenesis. Analysed and discussed all data.; NC, Performd IP/blot experiments, subcloning and site directed mutagenesis. Analysed and discussed all data.; AFS, Performed IP/blot experiments. Performed subcloning and site directed mutagenesis. Performed phosphatase and kinase assays. Analysed and discussed all data.; DG, Made initial observations on the Fcp1-Gwl interaction and designed experiments. Performed phosphatase and kinase assays. Analysed and discussed all data. Conceived and wrote the manuscript, Conception and design, Acquisition of data, Analysis and interpretation of data, Drafting or revising the article.

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
