## [Decision Letter]

Thank you for submitting your work entitled "Fcp1 controls Greatwall kinase to promote PP2A-B55 activation and mitotic progression" for peer review at *eLife*. Your submission has been favorably evaluated by Tony Hunter (Senior Editor) and three reviewers.

The reviewers have discussed the reviews with one another and the editor has drafted this decision to help you prepare a revised submission.

The authors have previously reported that the Fcp1, a metal-dependent RNA polymerase CTD phosphatase, is required for dephosphorylation of CDK1 substrates and exit from mitosis in mammalian cells. Here, they provide evidence that Fcp1 mediates dephosphorylation of CDK1 targets indirectly by dephosphorylating two CDK1 sites on Greatwall (Gwl), a kinase that through phosphorylation of endosulfine creates a potent inhibitor of PP2A/B55, the phosphatase that is responsible for dephosphorylating the majority of the CDK1 substrates, an event needed for exit from mitosis. In summary, here the authors propose a new link between Fcp1 and the Greatwall/Ensa/B55 pathway and suggest that Fcp1 lies at the top of this cascade and triggers mitotic exit by dephosphorylating and inactivating Greatwall kinase.

The reviewers each found strength in the paper, but were not fully convinced that your data completely justify your conclusions, and find that several aspects of your model have to be tested more rigorously. The following major points will need to be addressed in a revised version.

Essential revisions:

1) The authors' model places Fcp1 at the top of the mitotic exit cascade and one would expect a strong mitotic phenotype in Fcp1 knockdown cells, and certainly at least a significant delay in mitotic exit and an increase in mitotic index and/or polyploidy related to cytokinesis problems. Most of the experiments that support a role for Fcp1 were done using nocodazole release, and there is only a single picture shown of a binucleated cell without any quantification of the Fcp1 mitotic phenotypes. It is essential to show quantitative data that Fcp1 knockdown in asynchronous cells causes a significant increase in mitotic index and delays progression from M-phase into G1 phase resulting in binucleation to support the idea that Fcp1 is the key to this cell cycle transition. This is especially important since other siRNA screens (e.g. mitocheck) have not picked up any mitotic phenotypes with Fcp1.

2) The experiments in Figure 1 show clearly that Fcp1 depletion caused a block in cyclin B degradation in cells released from nocodazole. Thus, most of the subsequent data on Greatwall dephosphorylation that were carried out using the nocodazole-release synchronization method could simply be a consequence of Cdk1 activity remaining high because of cyclin B stabilization. The authors did carry out one set of experiments using the RO3306 Cdk inhibitor (Figure 1), but Gwl phosphorylation was not assessed here, and the authors relied on measuring downstream events. The effect of RO3306 Cdk inhibition on Gwl phosphorylation should be tested.

3) The authors failed to cite the Hégarat paper (PLoS Genet. 10:e1004004, 2014), which showed that Fcp1 depletion caused a block in Ensa dephosphorylation, but did not affect Gwl T-loop dephosphorylation. Are the conclusions of the authors' study compatible with those in the Hégarat et al. paper? It is essential to discuss this point (e.g. are there experimental differences?) to avoid confusing readers. In this regard, there is a critical experiment in the Hégarat paper that the authors should take into account: in Okadaic Acid treated cells, that are forced to exit mitosis by Cdk1 inhibition, Gwl retains approximately 60% of its activity as measured by IP kinase assays using Gwl purified before and after mitotic release. This would suggest that an OA-sensitive phosphatase is the major Gwl phosphatase at relevant activating phosphosites. The authors should perform a similar experiment in which they purify Gwl from Fcp1-depleted mitotic cells before and after Cdk1 inhibition, and measure its kinase activity in vitro. This would definitively determine how much Fcp1 does indeed contribute to Greatwall inactivation during mitotic exit.

4) Given the alternative hypothesis that PP1 rather than Fcp1 is responsible for Gwl inactivation, the Okadaic Acid experiments shown in Figure 2 and Figure 1—figure supplement 1 take on an added significance. In Figure 2, the dephosphorylation of pS90 and pS453 was not blocked by OA, implying that these dephosphorylations are catalyzed by Fcp1, a phosphatase not sensitive to OA. However, the concentration of OA used in these experiments (500 nM) is uncomfortably close to the borderline. PP2A-like enzymes are almost certainly nearly completely inhibited by this dose of OA, but the drug binds significantly less tightly to the active site of PP1-like enzymes. For complete inhibition of all the PP1-like enzymes, the stoichiometric concentration of the inhibitor must be in excess of the intracellular concentration of all proteins that bind it with K_i_ values equal to or lower than the dissociation constant for PP1. In other words, it remains possible that 500 nM of Okadaic Acid is sufficient to achieve quantitative inhibition of PP2A but not of PP1. Thus, the experiment done in Figure 2 needs to be repeated with a higher dose of OA, preferably 2 micromolar or above, but any type of evidence that PPI is not involved would suffice.

5) Figure 2 and part of Figure 3: In the experiments evaluating Fcp1's role in cells, the authors expressed and compared Fcp1-WT and Fcp1-CD (catalytically-dead mutant). Ideally, however, they should compare cells siRNA-depleted of endogenous Fcp1 with mock-treated cells (and possibly siRNA-depleted cells re-expressing siRNA-resistant WT), because it is difficult to control expression level of an exogenous gene similar to the endogenous one (as shown in Figure 1), and because the transfection efficiency of an exogenous gene never reaches 100%, making the cell population more heterogeneous. In addition, exogenous overexpression of a gene can cause artificial effects that cannot be predicted. As the authors have already done this in Figure 1, it is technically doable. To support the main conclusion it would be important that Figure 2 and the bottom half of Figure 3 (at least for pS67-Ensa) are re-tested in this context (mock/Fcp1-knockdown/rescued).

---

## [Author Response]

Essential revisions:

1) The authors' model places Fcp1 at the top of the mitotic exit cascade and one would expect a strong mitotic phenotype in Fcp1 knockdown cells, and certainly at least a significant delay in mitotic exit and an increase in mitotic index and/or polyploidy related to cytokinesis problems. Most of the experiments that support a role for Fcp1 were done using nocodazole release, and there is only a single picture shown of a binucleated cell without any quantification of the Fcp1 mitotic phenotypes. It is essential to show quantitative data that Fcp1 knockdown in asynchronous cells causes a significant increase in mitotic index and delays progression from M-phase into G1 phase resulting in binucleation to support the idea that Fcp1 is the key to this cell cycle transition. This is especially important since other siRNA screens (e.g. mitocheck) have not picked up any mitotic phenotypes with Fcp1.

Quantitative data of the phenotypes induced by Fcp1 knockdown in asynchronous cells, that show a significant increase in mitotic index, delayed progression through M-phase, binucleation and multinucleation in addition to cell death, have now been provided in Figure 1—figure supplement 4. The reason why these very evident phenotypes induced by Fcp1 downregulation, in our hands, were not picked up in previous large siRNA screens (e.g. mitocheck) is obscure to us. It is worth noting, however, that while the previously mentioned large siRNA screens did not score any phenotype upon Fcp1 downregulation, in other very recent large genetic knockdown screens, Fcp1 was classified as an essential gene in human cells (Science 27 November 2015: 1092-1096 and: 1096-1101. Published online 15 October 2015). Perhaps large screens may sometimes mislead to inaccurate conclusions.

2) The experiments in Figure 1 show clearly that Fcp1 depletion caused a block in cyclin B degradation in cells released from nocodazole. Thus, most of the subsequent data on Greatwall dephosphorylation that were carried out using the nocodazole-release synchronization method could simply be a consequence of Cdk1 activity remaining high because of cyclin B stabilization. The authors did carry out one set of experiments using the RO3306 Cdk inhibitor (Figure 1), but Gwl phosphorylation was not assessed here, and the authors relied on measuring downstream events. The effect of RO3306 Cdk inhibition on Gwl phosphorylation should be tested.

The experiments suggested have now been performed, and the effects of Cdk1 inhibitor RO3306 on Gwl phosphorylation in control and Fcp1-depleted cells are now shown in Figure 2 and Figure 2—figure supplement 2. In addition, previous Figure 1 has now been removed from the main text and is shown as Figure 1—figure supplement 1. Moreover, following the objection expressed in Essential Revision point 5, we have removed data concerning experiments involving re-expression of Fcp1 catalytic dead (Fcp1 CD) in Fcp1-siRNAs-treated cells.Previous Figure 1 were changed to Figure 1.

3) The authors failed to cite the Hégaratpaper (PLoS Genet. 10:e1004004, 2014), which showed that Fcp1 depletion caused a block in Ensa dephosphorylation, but did not affect Gwl T-loop dephosphorylation. Are the conclusions of the authors' study compatible with those in the Hégarat et al. paper? It is essential to discuss this point (e.g. are there experimental differences?) to avoid confusing readers. In this regard, there is a critical experiment in the Hégarat paper that the authors should take into account: in Okadaic Acid treated cells, that are forced to exit mitosis by Cdk1 inhibition, Gwl retains approximately 60% of its activity as measured by IP kinase assays using Gwl purified before and after mitotic release. This would suggest that an OA-sensitive phosphatase is the major Gwl phosphatase at relevant activating phosphosites. The authors should perform a similar experiment in which they purify Gwl from Fcp1-depleted mitotic cells before and after Cdk1 inhibition, and measure its kinase activity in vitro. This would definitively determine how much Fcp1 does indeed contribute to Greatwall inactivation during mitotic exit.

The Hégarat paper has now been discussed and cited (Results and Discussion, fourth and last paragraphs). In addition, the experiments suggested to measure activity of Gwl isolated from control and Fcp1-depleted cells before and after Cdk1 inhibition have been performed, and the results are shown in Figure 4 and Figure 4—figure supplement 2. Furthermore, we have included in this revised version a set of data that we already had, but did not include in the previous version, showing that active Fcp1 is not able to dephosphorylate pS67-Ensa in vitro under our experimental conditions, now shown in Figure 2. Since a paper showing the relevance of PP1 in Gwl inactivation was published during the preparation of this revised manuscript (Heim et al., 2015), we have discussed and cited that paper too and performed experiments related to PP1 under conditions of Fcp1 depletion and Cdk1 inhibition, now shown in Figure 2 and Figure 4. We tried to homogeneously integrate all previous information with ours (please see the last paragraph of the Results and Discussion).

4) Given the alternative hypothesis that PP1 rather than Fcp1 is responsible for Gwl inactivation, the Okadaic Acid experiments shown in Figure 2 and Figure 1—figure supplement 1 take on an added significance. In Figure 2, the dephosphorylation of pS90 and pS453 was not blocked by OA, implying that these dephosphorylations are catalyzed by Fcp1, a phosphatase not sensitive to OA. However, the concentration of OA used in these experiments (500 nM) is uncomfortably close to the borderline. PP2A-like enzymes are almost certainly nearly completely inhibited by this dose of OA, but the drug binds significantly less tightly to the active site of PP1-like enzymes. For complete inhibition of all the PP1-like enzymes, the stoichiometric concentration of the inhibitor must be in excess of the intracellular concentration of all proteins that bind it with K_i_ values equal to or lower than the dissociation constant for PP1. In other words, it remains possible that 500 nM of Okadaic Acid is sufficient to achieve quantitative inhibition of PP2A but not of PP1. Thus, the experiment done in Figure 2 needs to be repeated with a higher dose of OA, preferably 2 micromolar or above, but any type of evidence that PPI is not involved would suffice.

The experiment shown in previous Figure 2 has been repeated with a higher dose of OA, 2 μM, as recommended. The results are now shown in new Figure 2. In addition, new Figure 2, lower panels, shows that at this concentration OA is able to completely prevent auto-dephosphorylation of PP1 catalytic subunit alpha at T320, the inhibitory Cdk1-phosphorylated site in mitosis. These data indicate that this concentration of OA is indeed able to bind tightly to the active site of PP1 and inhibit the enzyme but, despite maintenance of Cdk1-dependent, inhibitory, phosphorylation of PP1, it is not able to prevent Gwl dephosphorylation at pS90 and pS453. Previous Figure 1—figure supplement 1, now Figure 1—figure supplement 2, was not changed since OA was used at 500 nM to essentially inhibit PP2A action and as you mentioned “PP2A-like enzymes are almost certainly nearly completely inhibited by this dose of OA*”*.

5) Figure 2 and part of Figure 3: In the experiments evaluating Fcp1's role in cells, the authors expressed and compared Fcp1-WT and Fcp1-CD (catalytically-dead mutant). Ideally, however, they should compare cells siRNA-depleted of endogenous Fcp1 with mock-treated cells (and possibly siRNA-depleted cells re-expressing siRNA-resistant WT), because it is difficult to control expression level of an exogenous gene similar to the endogenous one (as shown in Figure 1), and because the transfection efficiency of an exogenous gene never reaches 100%, making the cell population more heterogeneous. In addition, exogenous overexpression of a gene can cause artificial effects that cannot be predicted. As the authors have already done this in Figure 1, it is technically doable. To support the main conclusion it would be important that Figure 2 and the bottom half of Figure 3 (at least for pS67-Ensa) are re-tested in this context (mock/Fcp1-knockdown/rescued).

Figure 2 and Figure 3 have been changed following the suggested experiments and recommendations. Gwl and Ensa dephosphorylation have been compared in experiments with cells siRNA-depleted of endogenous Fcp1, mock-treated and siRNA-depleted re-expressing siRNA-resistant Fcp1 WT in new Figure 2. Following the reviewers’ suggestions, data concerning re-expression of Fcp1CD have also been removed from present Figure 3, in addition to present Figure 1 and Figure 1—figure supplement 1.